# Nipah Virus: A Multidimensional Update

**DOI:** 10.3390/v16020179

**Published:** 2024-01-25

**Authors:** Javier Faus-Cotino, Gabriel Reina, Javier Pueyo

**Affiliations:** 1School of Medicine, Universidad de Navarra, 31008 Pamplona, Spain; jfaus@alumni.unav.es; 2Microbiology Department, Clínica Universidad de Navarra, 31008 Pamplona, Spain; 3IdiSNA, Navarra Institute for Health Research, 31008 Pamplona, Spain; jpuevi@unav.es; 4Department of Anesthesia and Intensive Care, Clínica Universidad de Navarra, 31008 Pamplona, Spain

**Keywords:** Nipah virus, outbreak, zoonosis, tropical disease, emerging pathogen, Hendra virus

## Abstract

Nipah virus (NiV) is an emerging zoonotic paramyxovirus to which is attributed numerous high mortality outbreaks in South and South-East Asia; Bangladesh’s Nipah belt accounts for the vast majority of human outbreaks, reporting regular viral emergency events. The natural reservoir of NiV is the Pteropus bat species, which covers a wide geographical distribution extending over Asia, Oceania, and Africa. Occasionally, human outbreaks have required the presence of an intermediate amplification mammal host between bat and humans. However, in Bangladesh, the viral transmission occurs directly from bat to human mainly by ingestion of contaminated fresh date palm sap. Human infection manifests as a rapidly progressive encephalitis accounting for extremely high mortality rates. Despite that, no therapeutic agents or vaccines have been approved for human use. An updated review of the main NiV infection determinants and current potential therapeutic and preventive strategies is exposed.

## 1. Introduction

Emerging infectious diseases represent a large threat for global health. Their emergence points to novel pathogenic agents that frequently have a zoonotic origin, which encourages the tight link with human activity in terms of natural environment modification and close human–wildlife contact, and to the arising of previously undiscovered viruses [1,2]. Nipah virus (NiV) is a high mortality zoonotic emergent pathogen, classified as Biosafety Level 4, which represents a considerable concern for international public health, in relation with the current lack of effective therapeutic or vaccinal agents approved for human use [3].

## 2. Nipah Virus Outbreaks

### 2.1. Malaysia-Singapore 1998–1999

NiV first emerged in Malaysia as a pig livestock disease transmitted to humans, causing a large encephalitic outbreak with an estimated mortality of 40%, which accounted for 276 cases and 106 deaths [4]. The epicenter was retrospectively located in a farm in the Ipoh village, where the first human case was reported in January 1997. Afterwards, the virus disseminated throughout Peninsular Malaysia and Singapore through livestock sales during 1998/1999 [5,6]. The farm generated a key intersection between commercial fruit production, the livestock industry, and the local circulation of *Pteropus* bats (natural viral reservoir), establishing an ideal scenario for an outbreak origin. Fruit became an attraction for *Pteropus*, enabling the contact of livestock with chewed fruits and bat excreta spilled into pigsties, leading to bat–pig transmission and viral circulation in a naïve pig population. Posterior pig-to-human transmission initiated the human outbreak [1,7]

### 2.2. Bangladesh 

Multiple outbreaks have occurred in the country since the first detection of NiV in 2001, which has placed the territory as the main source of reported human cases [4,8]. Several characteristics differentiate the outbreaks in Bangladesh compared with Malaysia-Singapore 1998–1999; the territory exhibits a noticeable reiteration of direct viral emergences from bats to human via the ingestion of contaminated date palm sap, without the requirement of an intermediate amplification host, which tends to repeat annually with a marked seasonal pattern (Nipah season) [9,10,11]. Clinical features show common respiratory involvement, which relates to the interhuman airway transmission observed, and a median mortality rate of 75% [5,9].

### 2.3. India

The territory reported outbreaks in 2001, 2007, and 2018, showing similar patterns of interhuman airway transmission as in Bangladesh [9,12].

### 2.4. Philippines

During 2014, an outbreak occurred involving equine livestock, in which horse-to-human, interhuman, and food transmission was reported. Food transmission was directly attributed to horse meat consumption [4].

## 3. Nipah Virus Characterization

NiV is an enveloped negative sense single-strand RNA virus belonging to the *Paramyxoviridae* family, *Paramyxovirinae* subfamily, *Henipavirus* genus. “Nipah” refers to Sungai Nipah Village, Malaysia, the area where the agent was first identified. The virus is closely related to Hendra virus (HeV); both of them manifest biological, serological, and molecular properties that differ from other Paramyxoviruses, requiring the creation of a new genus, *Henipavirus* (HNV). In recent times, other novel species have joined the HNV genus, such as Cedar, Mojiang, and Ghana bat viruses [3,13].

### 3.1. Viral Structure

A pleomorphic extracellular bilayer defines the outer viral surface in which the attachment (G) and fusion (F) proteins are inserted. The inner surface is shaped by matrix proteins (M) constituting the viral capsid, which contains a single RNA strand attached to nucleocapsid viral proteins (N), phosphoproteins (P), and the large RNA polymerase enzyme (L) [14]. The viral genome comprises six genes (N, P, M, F, G, L), which codify the six major structural proteins as shown in Figure 1. Post-transcriptional editing and alternative start codons allow the processing of alternative protein products (P, V, W, C) from the P gene. Proteins V and W are capable of inhibiting Interferon production by mammalian cells, whereas P, V, and W interfere with Interferon cellular signaling. Protein C typically locates in host cell cytoplasm, mediating viral budding and release of new virions [14,15,16].

### 3.2. Phylogeny and Viral Strains 

The wide distribution of the HNV genus and the scarce observation of infectious symptomatology in *Pteropus* bats suggest that NiV has most likely co-evolved with the *Pteropus* genus, establishing a long-standing relationship which accounts for the diminished pathogenic capacity over its natural reservoir [18]. Multiple viral isolations demonstrate the existence of at least two genetically differentiated viral strains, NiV-M (Malaysia) and NiV-B (Bangladesh), whose features may partially explain the variation of lethality, infectivity, and pathogeny observed in human cases during outbreaks [19,20]. Whereas all human cases reported during the Malaysia-Singapore outbreak point to a single viral variant belonging to NiV-M, the multiple outbreaks occurring in Bangladesh and India did originate from a number of characteristic viral variants belonging to NiV-B, which underlines the multiple viral emergences from the natural reservoir (*Pteropus* spp.) to humans [21].

## 4. Nipah Virus Reservoir and Host Range

### 4.1. Natural Reservoir 

Multiple viral isolations, as well as serological evidence, point to *Pteropus* spp. as the natural reservoir, including multiple species such as *P. hypomelanus*, *P. vampyrus*, *P. lylei*, *P. giganteus*, etc. [7,22]. *Pteropus* genus distribution includes Africa, South and South-East Asia, Australia, and a number of South Pacific islands, and either NiV viral isolation or serological evidence of HNV infection has been noticed in every place where *Pteropus* spp. have been studied (Malaysia, Indonesia, Cambodia, Bangladesh, India, Thailand, Myanmar, Australia, Papua New Guinea, Madagascar, etc.) [4,18,23]. The vast range comprising *Pteropus* spp. overlaps all Nipah outbreak locations, but the Nipah belt clearly stands out as the most affected territory, which covers the central and northwestern regions of Bangladesh [5,24].

Infected bats disseminate viruses by natural secretions and excreta (saliva, urine, seminal fluid, feces, etc.), and, when exposed to contaminated materials, they generally seroconvert. Seroconversion is incapable of preventing new infections; therefore, reinfection leads to the persistence of viral circulation within *Pteropus* populations [21,22].

### 4.2. Intermediate Amplifier Hosts

NiV has an unusually wide range of possible amplification hosts, extending over six mammal orders, including human, pig, dog, cat, horse, hamster, guinea pig, bat, and ferret. Both pigs and horses have played key roles as viral amplifier hosts in Malaysia-Singapore and Philippines outbreaks, respectively [3,6,13].

## 5. Nipah Virus Infection

### 5.1. Pathogeny

The virus reaches the respiratory tract following aerosol transmission and respiratory droplets. After situating in upper and lower respiratory epithelium, virus particles penetrate epithelial cells by a membrane fusion process. The tetrameric G attachment glycoprotein head region interacts with cellular surface Ephrin B2/B3 receptors; simultaneously, the C-terminal portion of G glycoprotein activates the trimeric F fusion glycoprotein. The F protein contains two hydrophobic domains which form a six-helical bundle able to penetrate the host cell membrane, thus initiating membrane fusion and posterior viral replication using host cell machinery [14,25,26]. The expression of F and G viral glycoproteins in basolateral host cell membranes allows local dissemination by promoting membrane cellular fusion with adjacent epithelial cells and formation of characteristic cellular syncytia [27,28] (Figure 2). Systemic dissemination takes place by viral budding and release which determines extension to mucosal-associated lymphoid tissue (MALT) and leucocyte attachment, leading the virus to reach the blood stream [14].

### 5.2. Viral Immune Evasion

As NiV and the *Pteropus* spp. have most likely co-evolved for centuries, it is assumed that NiV has developed specific evasion mechanisms against *Pteropus* immune response. Therefore, the high interspecies immunity conservation observed in mammals may partially explain the wide infectious range of the virus. NiV blocks innate and adaptive immune responses by the inhibition of synthesis and release of cytokines, mainly IFN type 1 α/β, as well as CCL4, CCL5, and TNF-α. Among all P gene-derived products, the major inhibitory capacity remains on the W protein, which blocks IFN type 1 α/β gene expression from host cell nuclei [14].

### 5.3. Viral Tropism and Tissular Lesion

Both viral tropism and tissular lesions are determined by the expression of Ephrin B2/B3, which acts as a G attachment protein receptor. Ephrin B2/B3 are membrane proteins related to neuronal development, cellular migration, and vasculogenesis, and are mostly expressed in the vascular endothelium and central nervous system. Tissular lesions express as wide distribution ischemia derived from vasculitis and thrombosis phenomena, including diffuse nervous tissue lesions and formation of multinucleated giant cells in vascular endothelium. At the pulmonary level, alveolar hemorrhage, pulmonary edema, and parenchyma and small vessel affection are typical findings that correlate with the natural progression to interstitial pneumonia and acute respiratory distress syndrome (ARDS) [9,13,21,29].

### 5.4. Clinical Presentation

Acute infection develops as sudden-onset rapidly progressive encephalitis, frequently associated with respiratory involvement as atypical pneumonia and disseminated vasculitis. A flu-like syndrome occurs after the incubation period (estimated 7–11 days), which progresses with moderate to high-grade fever, altered cognition, shortness of breath, and cough. The subsequent neurologic and respiratory deterioration, including encephalitis and ARDS, leads to multisystemic failure and eventual death, typically due to brainstem involvement (mean of six days between onset of symptoms and death). A 75% mortality rate is estimated, and asymptomatic cases are infrequent; 22% of survivors develop residual neurological deficits. A noticeable differentiation of clinical presentation was found between the Malaysia-Singapore outbreak, which exhibited an absence of respiratory involvement, and the posterior Bangladesh and India outbreaks, where respiratory conditions are a common feature [18,19,20,21,30,31].

Occasionally, a late-onset or recurring presentation has been observed, which may develop several months after the initial exposure to NiV, and generally shows confined CNS effects. Viral persistence mechanisms remain unknown. A lower mortality rate (18%) but an increased risk of residual neurologic deficits (61%) has been documented for this form of presentation [9,30,32].

## 6. Nipah Virus Transmission

### 6.1. Interspecies Transmission

NiV transmission from *Pteropus* spp. to humans requires an interaction interface which allows viral emergence from its natural reservoir. The most important one is fresh palm sap, as well as consumption of derived products, which accounts for the vast majority of outbreaks in Bangladesh—the only country reporting annual viral emergences [23]. Date palm sap is a seasonal national delicacy, whereas tari is its fermentation product. The sap is harvested from palm trees at night, poured into a bigger receptacle, and sold during the early morning prior to fermentation and sweet taste loss. This procedure favors virus load survival and dissemination by persistence in the contaminated sap [4,8,33]. Much evidence points to frequent sap consumption by *Pteropus* during the harvesting process (Figure 3), enabling sap contamination by viral dissemination in saliva, urine, or feces. Depending on environmental conditions, NiV could survive for four days in bat urine and 24 h in date palm sap, features that place the sap as an adequate viral vehicle [8,23,33].

Nipah outbreaks in Bangladesh follow a noticeable seasonal pattern which overlaps with date palm sap harvesting season. Most outbreaks have been recorded during the coldest months of the year, always between the January–May window. Additionally, during this time of the year, the bat population becomes stressed by the scarcity of natural food sources, making *Pteropus* spp. move closer to the villages. It has also been hypothesized that the adverse environmental conditions of winter may weaken the immune system of the *Pteropus*, favoring an increased viral load and dissemination during this period [21,34].

Other interspecies transmission mechanisms observed include airway transmission from pigs to humans and from horses to humans as documented in the Malaysia-Singapore and Philippines outbreaks, respectively, as well as food-borne transmission observed in the Philippines outbreak. Occasional Bangladesh outbreaks have been related to close contact with sick cows as well as activities related to exposure to bat secretions or excreta—for instance, while climbing trees [30,33].

**Figure 3 viruses-16-00179-f003:**
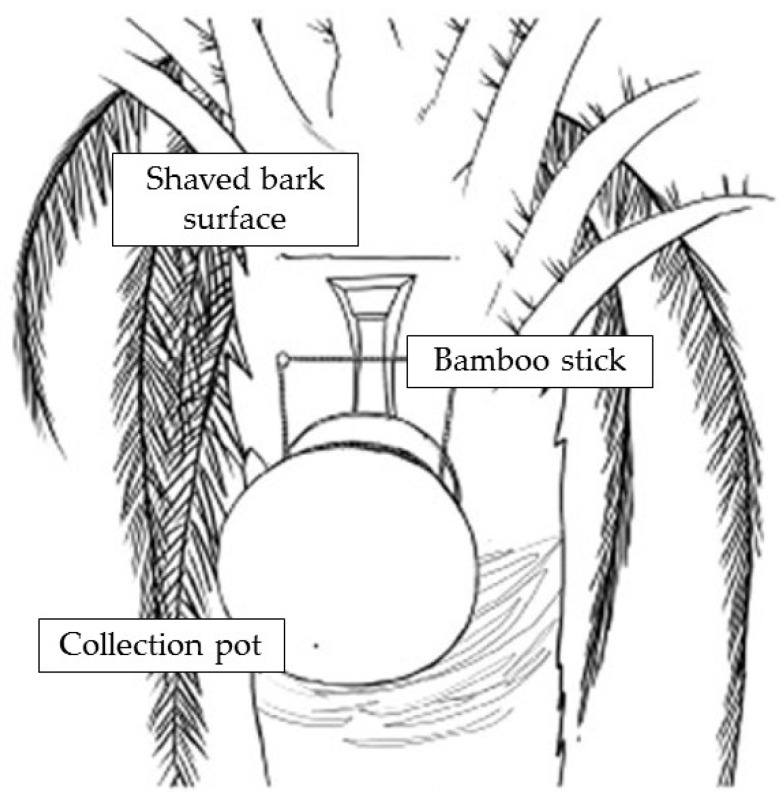
Date palm sap harvesting system; the palm’s bark is shaved close to the treetop; during the nighttime, the fresh sap flows into the receptacle [Adapted from [35]].

### 6.2. Human-to-Human Transmission 

During the Malaysia-Singapore outbreak, only pig-to-human transmission was documented, whereas interhuman transmission has been a common finding during the outbreaks located in Bangladesh and India, as shown in Figure 4. These differences might relate to respiratory involvement and the subsequent capacity of viral dissemination, viral strain properties or genotypes, patient basal health status, differences in health care attention, and other unknown variables [4,34]. In Bangladesh and India, contagion occurs mainly by airway transmission (via aerosols and droplets), and generally close physical contact with the infectious patient or their secretions is required for infection acquisition. In these countries, approximately one half of cases account for interhuman transmission, which is frequently linked to “super spreaders”, a subgroup of critically ill patients older than 25, who express dominant respiratory involvement and high viral load, as well as extensive NiV dissemination detected in respiratory secretions, saliva, and urine [13,21,36]. Despite these findings, outbreak analysis points to an R_0_ = 0.48, defining a low potential for self-sustaining viral transmission in human populations [18,19,37].

## 7. Countermeasure Requirements, Epidemic/Pandemic Potential of Nipah Virus and Public Health Implications

### 7.1. Implications of Bangladesh’s Outbreaks

Bangladesh is the only country reporting regular NiV outbreaks with a remarkable seasonal frequency, which repeats annually during wintertime. Between 2007 and 2014, 79 viral emergences have been reported in the country, with some of them accounting for 100% mortality rates. In addition to the habitual presence of interhuman transmission, Bangladesh’s outbreaks have a higher mortality and severity than Malaysia-Singapore’s. The frequency and severity of Bangladesh’s outbreaks have placed central and northern districts of the territory in a public health emergency [6,19,21,38]. Despite millions of people consuming fresh palm sap yearly in the area known as Nipah belt and the extraordinary virulence of NiV, so far, no therapeutic agents or vaccines are available for human use [23,39].

### 7.2. Large-Scale Implications

NiV is considered an emerging viral agent with characteristic epidemic/pandemic potential. Since 2017, it has been repeatedly included in the World Health Organization (WHO) list of priority agents requiring investment for the development of diagnostic, therapeutic, and preventive countermeasures, and the Coalition for Epidemic Preparedness Innovations (CEPI) includes NiV in the priority list of viral agents requiring the development of vaccination preventive measures [19,24,40]. The Centers for Disease Control and Prevention (CDC) and the National Institute of Allergy and Infectious Diseases (NIAID) consider Nipah virus a category C priority pathogen (potential biological weapon for bioterrorism) [21].

The main threat concerning global health is the emergence or selection of a more transmissible NiV strain. The observed capacity of transmission between patients and caregivers has raised the alert of a possible viral adaptation to a more effective human transmission. As an RNA virus, NiV possesses a high mutation rate, which, together with its wide host range and demonstrated capacity of interspecies and interhuman transmission, enables the selection of more transmissible strains as a component of the virus–host adaptation process. Because human infection entails a limitation for viral survival and dissemination, natural selection favors the emergence and dominance of new viral variants able to expand the infectious range so as to confront the adversity generated by the new environment [7,18].

Moreover, the emergence of more transmissible strains currently circulating in the natural reservoir remains plausible if the frequent bat-to-human transmission via palm sap consumption continues occurring in Bangladesh. Furthermore, this possibility extends to vast regions of Asia, Africa and Oceania, where the consumption of local date palm sap-derived beverages also overlaps with *Pteropus* spp. geographic distribution [4,8,23].

Zoonotic diseases can be divided into four stages, which include: I exclusive to animals;II primary human infections only;III limited interhuman transmission;IV sustained interhuman transmission.

The transition between stages III and IV occurs when R_0_ > 1, allowing the sustained establishment of the pathogen in a human population. NiV human infection belongs to stage III, placing the pathogen as a potential source of epidemics or pandemics. In addition to the established international transport connection and the high mortality rate of Nipah virus, the emergence of an R_0_ > 1 in the current context of the huge human density population settled in South Asia would be the origin of a devastating global pandemic [18,31,38]. More than two million people inhabit NiV outbreak risk regions, including some of the most densely populated territories on Earth; the transmission of a more transmissible pathogen in an overpopulated country with scarce health resources, such as Bangladesh, would likely allow wide viral dissemination prior to the instauration of effective infection control measures, leading to a global health emergency [4,13,34,36].

## 8. Therapeutic Agents

Current patient management is limited to supportive measures and palliative care as no therapeutic agents are approved for human use [21]. The development of novel therapeutic drugs should be a targeted future alternative; the most promising experimental agents are noted below and summarized in Table 1.

### 8.1. Nucleoside/Nucleotide Analogs

**Remdesivir**: broad-spectrum RNA polymerase inhibitor antiviral prodrug. It has demonstrated potent replication-inhibitory activity in in vitro and in vivo experimentation. It constitutes a potential therapeutic candidate and a possible viral clearance adjuvant for survivors as a recurrence preventive measure. High seropositive maintained titers relate to incomplete viral clearance in patients and increased risk of infection recurrence; the intense viral clearance observed in experimental animals treated with remdesivir supports its use on selected patients [40].**Favipiravir**: selective RNA polymerase inhibitor antiviral drug. Great replication- and transcription-inhibitory capacity observed in vitro even with low drug concentrations. Experimental animal models with Golden Hamster reveal complete protection against lethal NiV doses. Considered as a highly potential human therapeutic agent as well as a post-exposure prophylaxis candidate [39].**Ribavirin**: broad-spectrum antiviral used in Malaysia-Singapore outbreak alone or combined with Chloroquine, showing a 36% relative mortality risk reduction. However, in vivo experimentation did not support any mortality reduction including African Green Monkey animal models [6,9,39].**Balapiravir**: it has demonstrated in vitro efficacy against NiV, currently pending in vivo experimentation [39].

### 8.2. Monoclonal Antibodies

**m102.4**: a highly potent and promising human monoclonal antibody directed to the G viral glycoprotein–Ephrin B2/B3 interaction surface. For its synthesis, it has been selected by the screening of G-soluble glycoprotein form antibody library and exposed to a posterior affinity maturation by light chain reconstruction and random mutation of heavy chain variable regions [40,41]. It exhibits extensive neutralizing in vitro activity against all tested NiV and HeV strains. In vivo efficacy against NiV lethal doses has been confirmed in African Green Monkey and ferret experimental animal models in therapeutic conditions within several post-viral inoculation drug administration intervals (including after the initiation of clinical signs and circulating virus detection). It maintains an appropriate in vivo stability, and its biological activity extends until eight days post-infusion in animal models. Since 2010, m102.4 has been employed in humans for compassionate use for post-exposure treatment against NiV and HeV in USA, India, and Australia; however, real human efficacy remains unknown [6,39,42].**h5B3.1**: a humanized monoclonal antibody directed to the prefusion conformation of F viral glycoprotein; able to prevent virus–host cell membrane attachment and viral penetration by junction with quaternary F glycoprotein epitopes. Considered as a potential candidate for post-exposure treatment/prophylaxis. Currently pending in vivo experimentation [42].**nAH1.3**: a broadly neutralizing antibody interfering with the F fusion-triggering mechanism and a potency comparable to that of m102.4. They both have distinct antigenic sites that are non-competitive and could be a potential treatment option in combination [43].

Additional monoclonal antibodies for various epitopes have been developed and show a high potential for HNV neutralization by inhibiting binding to host cells: h1F5, h12B2, Nip GIP35, Nip GIP1.7, HENV26, HENV32, 4H3, 2D3, 1H8, 1F2, 1F3, 4B8, 1A9, 2B12, and 4F6. These agents must be evaluated in vivo in the near future to test for safety and efficacy and to study their potential clinical use in the future [44].

### 8.3. Fusion-Inhibitory Peptides

**Cholesterol-tagged fusion-inhibitory peptides**: designed to block F glycoprotein conformational changes leading to viral pore-mediated host cell penetration. Cholesterol tag provides a drastic increase in antiviral efficacy by directing peptides to target host cell membrane. Golden Hamster experimental models reveal an 80% survival rate with lethal viral doses when administered at the same time as viral inoculation, showing elevated penetration and high drug concentration in CNS, lung, and vascular endothelium. A remarkable decrease in survival rate (40%) has been observed when infusion is delayed by 48 h [45].**Inhaled fusion-inhibitory lipopeptides**: as an advantage, the inhaled administration route allows focused respiratory system delivery, covering the main viral entry pathway. In vivo experimentation has been conducted in Golden Hamster and African Green Monkey, a 33% relative mortality risk reduction was observed for the African Green Monkey model [40,46].

Nanoparticle formulation remains as a promising future alternative for therapy, but no in vivo or in vitro experiments have been conducted to date [14]. 

**Table 1 viruses-16-00179-t001:** Comparison of efficacy, availability, safety and advantages of therapeutic agents against Nipah Virus [6,9,39,40,41,42,45,46].

Drug	Efficacy	Endemic Region Availability	Safety	Advantages
Remdesivir	Broad-spectrum antiviral, in vitro efficacy against NiV-M and NiV-B. In vivo efficacy in African Green Monkey animal model against NiV-B. Unable to prevent local viral replication in respiratory tract. No in vivo efficacy tested against NiV-M	As a small-molecule antiviral agent, it exhibits greater facilities for storage and distribution compared with monoclonal antibodies and peptide-based strategies.	Previous clinical trials conducted in humans for other purposes. Experience in human use against other viruses (COVID-19, Ebola).	Potential use as viral clearance adjuvant and recurrent infection preventive strategy. Considered an adequate adjuvant for combined drug therapy against NiV.
Favipiravir	Broad-spectrum antiviral efficacy, including in vitro efficacy against NiV-M. In vivo efficacy in Golden Hamster animal model against NiV-M. No in vivo efficacy tested within other time-to-treatment intervals. No in vivo efficacy tested against NiV-B.	As a small-molecule antiviral agent, it exhibits greater facilities for storage and distribution compared with monoclonal antibodies and peptide-based strategies.	Human clinical trials conducted in humans for other purposes. Approved for human use against influenza virus in Japan. Phase 2 and 3 clinical trials against other viruses (COVID-19, Ebola).	Considered a potential candidate for post-exposure prophylaxis.
m102.4	High in vitro efficacy against NiV-M and NiV-B. High in vivo efficacy against NiV-M in ferret and African Green Monkey animal models. Adequate stability and activity retention for 8 days in animal model. No in vivo efficacy tested against NiV-B.	Limited availability for massive use and deployment to rural areas during outbreaks. Cold chain transportation and storage requirement. Intravenous route of administration.	Phase 1 human clinical trial initiated in 2019 (agent safety evaluation).	Considered the most promising candidate for human use. Exceptionally potent against NiV. Currently the only therapeutic agent showing efficacy in non-human primates within therapeutic administration conditions. Human tested for compassionate drug use in high-risk exposures to NiV/HeV individuals in India, USA, and Australia. Possible antiviral resistance development (in vitro observation) through viral mutation or new strain emergence.
Fusion-inhibitory peptides/lipopeptides	Inhaled fusion-inhibitory lipopeptides: in vivo efficacy in Golden Hamster and African Green Monkey animal models; adequate stability for inhaled route of administration. Cholesterol-tagged fusion-inhibitory peptides: in vivo efficacy in Golden Hamster animal model. Optimal penetration in NiV infection target organs.	Adequate rural area implementation potential regarding administration route properties and viability of rapid deployment during outbreaks.	No human clinical trials conducted.	Drug adjustment possibility against viral resistant strain emergence or viral mutation.

Efficiency and effectiveness criteria are not included due to lack of available information. Balapiravir, h5B3.1, and nanoparticle-based therapies have been discarded from the comparison attending to lack of in vivo experimentation trials. Ribavirin has been discarded from the comparison due to lack of mortality reduction observed in vivo.

## 9. Preventive and Outbreak Control Strategies

The availability of infection management and control guidelines, as well as definitions of case criteria, are considered a basic step to approach detection, management, and prevention activities. Bangladesh’s National Guideline for Management, Prevention, and Control of Nipah Virus Infection including Encephalitis has been published for this purpose, considering the determinants of Bangladesh’s outbreaks [21]. Different strategies for NiV prevention and control are summarized in Table 2.

### 9.1. Surveillance and Outbreak Detection Strategies

Considered crucial for early outbreak detection and containment. 

**Populational surveillance systems**: several strategies have been implemented previously in Bangladesh with this aim, including Nipah belt-focused hospital surveillance (plus Nipah season intensification), 24 h hotline implementation for adverse health events notification, and mass media information monitoring [21,47]. Mass media monitoring was implemented from 2010–2011 by using Bangladesh’s National Media-Based Public Surveillance System, and it represented a highly effective, low-cost, sustainable outbreak detection strategy, which suits low-income countries with scarce health infrastructure development, such as Bangladesh. It is based on the extensive monitoring of the principal media information sources for the early detection and investigation of possible NiV outbreaks [47].**Exposure-based screening**: the use of brief hospital admission questionnaires to assess patients’ previous risk exposure was demonstrated to be highly effective and efficient when implemented in Bangladesh during 2012–2013 for early NiV-encephalitis case detection and interhuman transmission prevention. Asking for sap consumption and history of contact with febrile patients with altered cognition during the previous 30 days since symptom onset proved to be a useful NiV screening tool, especially during Nipah season (observed negative predictive value NPV = 99% during wintertime). In addition, screening questionnaire implementation allows a more efficient use of NiV transmission prevention resources in hospital settings, which are frequently scarce [36].

### 9.2. Human Infection and Transmission Preventive Interventions

Outbreak investigation in Bangladesh revealed two main transmission routes: bat-to-human transmission through fresh sap consumption and interhuman transmission through close contact with an NiV-infected patient. Focusing on interhuman transmission, most contagions affect family members and care providers either by direct contact with the patient, corpse handling, or fomite transmission [48]. Therefore, the implementation of basic infection control principles capable of limiting direct patient and corporal fluid exposure are crucial for the prevention of interhuman transmission. These basics include hand washing, wearing masks and gloves, eye protection, and the use of personal protection equipment [21,49,50]. 

**Table 2 viruses-16-00179-t002:** Preventive strategies against Nipah Virus, comparison of efficacy, effectiveness, cost efficiency and endemic region availability [5,7,8,9,13,15,18,19,21,23,30,31,35,36,47,51].

Strategy	Efficacy	Effectiveness	Cost Efficiency	Endemic Region Availability	Comments
Populational surveillance systems	High efficacy for early NiV outbreak detection and health response.	Elevated effectiveness observed when *National Media-Based Surveillance System for Public Health Events* was implemented during 2010–2011 season.	Sustainable economic intervention for early outbreak detection in low-income developing countries, i.e., Bangladesh.	Available intervention; previously implemented in Bangladesh.	Health setting independence for possible outbreak notification in rural regions; avoids laboratory-dependent diagnosis delay for potential public health risk events. Avoids high costs derived from laboratory-dependent diagnosis for possible outbreak notification. Remains an effective surveillance tool for vast regions which are distant to health settings.
Exposure-based screening	Proven efficacy for early NiV-encephalitis case detection in hospital settings.	High effectiveness observed when implemented in hospital settings during 2012 and 2013.	Enables an efficient utilization of scarce existing resources for infection diagnosis and interhuman transmission prevention in hospital settings by discriminating possible NiV-encephalitis cases.	Available intervention; previously implemented in Bangladesh for hospital settings.	Allows a more adequate distribution of high costs derived from laboratory diagnostic tests, achieving more efficient resource management for possible NiV cases.
Populational/community educational intervention	Remarkable proven efficacy on high-risk exposure-related behavior modification in target populations.	High effectiveness observed when implemented in reduced scale during 2009. Broad-scale evaluation is required.	Interspecies transmission blocking is considered as the most efficient and cost-effective intervention for NiV outbreak prevention.	Available intervention; previously implemented in Bangladesh.	Educational interventions are considered essential for infection and interhuman transmission reduction. Possibility of message adequation attending to target populations. Current evidence suggest that modest reductions in fresh sap consumption can notably reduce the incidence of human outbreaks. Optimal functioning is not required to achieve extensive health impact in outbreak incidence.
Physical barrier bamboo skit method	Proven efficacy on bat-to-human transmission prevention by blocking *Pteropus*–sap physical contact and subsequent sap contamination. Considered as a high potential prevention candidate to significantly reduce human outbreaks by preventing interspecies transmission and outbreak onset.	Requires further effectiveness evaluation under habitual use conditions. No broad-scale evaluation has been conducted.	Considered as a low-cost intervention due to broad bamboo availability in rural Bangladesh and reuse potential. Interspecies transmission blocking is considered as the most efficient and cost-effective intervention for NiV outbreak prevention.	Available intervention; previously adopted by *gachhi* community in rural Bangladesh regions with different purposes.	Respects the socio-cultural tradition of fresh palm sap consumption in rural Bangladesh and *gachhi* collectivity derived from commercial activity. *Gachhis* consider *banas* as an acceptable intervention when asked, despite the extra resources and time required for its implementation. Considered an easy implementation strategy; installation only requires proper positioning anda surface covering for exposed sap. Can reduce the risk of NiV infection from both fresh sap and *tari* consumption.

Vaccine-based strategies have been discarded from the comparison due to limited experimental development (no vaccine candidate has been appoved for human use).

**Populational/community educational intervention**: a multilevel information campaign adjusted for specific risk factors in a targeted population (i.e., fresh sap consumption in rural areas, direct contact transmission in hospital settings, etc.) (Table 3). This strategy makes the most of educative measures to impact the prevention of infectious diseases [21]. During 2009, several messages were disseminated in rural Bangladesh to dissuade the population from sap consumption; the “only safe sap” campaign represented the best approach, achieving a great improvement in disease knowledge and transmission awareness. The message is a harm reduction approach that recognizes abstinence as an ideal outcome but accepts alternatives that reduce harm (“Only safe sap” refers to consumption of physical barrier harvested palm sap) [31].**Physical barrier strategies for bat-to-human transmission prevention**: the fresh date palm sap harvesting period overlaps the Nipah outbreak season as contaminated sap consumption is the main route for NiV emergence in Bangladesh. Local harvesters, known as “gachhis”, collect and sell fresh sap during the early morning, and consumers typically ingest the fresh sap within the same day. Under these conditions, viable viral load reduction in contaminated sap is minimal; hence, interventions that prevent sap contamination from *Pteropus* spp. Stand out as valuable strategies [21,31,51]. Traditionally “gachhis” have used several methods to prevent sap deterioration from bat urine and excrements, including the use of tree branches to cover the sap circulation area, bark impregnation with lime, and the placing of bamboo skirts. Infrared camera evidence has demonstrated that only bamboo skirts, locally known as “banas”, are capable of preventing bat–sap flow physical contact [35,51]. Camera evidence proves *Pteropus* spp. frequently visit date palm trees and contaminate the collecting sap flow by directly contacting and feeding from it (Figure 5). Although further experimentation is needed to assess whether the “bana” method offers large-scale effectiveness, it still remains as a high potential preventive strategy. Zoonotic transmission prevention is considered the most efficient and cost-effective strategy to prevent human outbreaks. Furthermore, it is a low-cost intervention, easy to install, and accepted by “gachhis” when asked, which allows the conservation of the traditional practice of fresh sap harvesting and consumption [13,35,51].

### 9.3. Vaccination

All human vaccine candidates remain in development, but no vaccines are approved for human use. 

**Subunit vaccines**: epitopes or viral peptides, highly specific, easy to produce low-cost vaccines. Bioinformatics tools predict the main epitopes are able to trigger a sufficient immune response; F and G glycoprotein fractions stand out as the best candidates [15]. Animal experimentation in African Green Monkey models reveals complete protection and development of high IgG levels against NiV with a subunit vaccine based on the oligomeric soluble form of G recombinant HeV glycoprotein (sGHeV). The elevated immunogenicity observed and its exceptional efficacy support future evaluation and, eventually, authorization for human use [5,9].**Vector vaccines**: attenuated virus able to express G/F NiV recombinant glycoproteins. Promising results have been observed in in vivo experimentation with pig and Golden Hamster animal models, showing adequate serological response with several viral vectors (canarypox virus, vesicular stomatitis virus, and Venezuelan equine encephalitis virus). Virus-like particles derived from mammal cells expressing F, G, and M viral proteins have also demonstrated the ability to induce a potent neutralizing response and complete protection against lethal doses of NiV in Golden Hamster models [9].**mRNA-1215 vaccine**: a candidate mRNA vaccine under evaluation in a Phase 1 clinical trial has been developed by NIAID and Moderna. It encodes for the prefusion state of the F protein covalently linked to the G protein monomer (pre-F/G) of the Malaysian strain NiV [52].

Passive immunization: a polyclonal serum against the G and F proteins remains as a potential candidate for this aim [9].

## 10. Comparison of Strategies Focused on Endemic Regions

In perfect conditions, any health intervention aiming to combat Nipah outbreaks in rural Bangladesh should accomplish a number of criteria that can guarantee key aspects, such as efficacy, safety, effectiveness, cost efficiency, and availability. Ideally, efficacy, safety, and prognosis impact evidence from interventions should come from human clinical trials which verify these aspects against both NiV-M and NiV-B strains, as well as including other parameters, such as pharmacokinetics, pharmacodynamics, administration route, posology, etc. [39,40]. As a high mortality pathogen, conventional human clinical trial evidence is limited; therefore, intervention efficacy evaluation is based on animal experimentation models able to recreate human disease. Non-human primates are preferable; currently the African Green Monkey (*Chlorocebus sabaeus*) is considered the best animal model due to its high clinical and pathological correlate with the human NiV infection [5,6,40]. Effectiveness verification should demonstrate adequate intervention functioning under usual conditions of use [53]. Decision analysis, economic evaluation, and cost efficiency verification require an adequate health plan comprising situation analysis, setting of objectives, generation of an activity plan, prediction of necessary resources, task organization, execution, and posterior evaluation, as well as favorable cost-effectiveness, cost–utility, and cost–benefit analyses [53,54]. In addition, attending to the existing conditions and epidemiological determinants of NiV outbreaks in rural Bangladesh, which count for scarce health infrastructure and little economic development, should be addressed [18]. The “availability to use criteria” requires a complete intervention development, as well as the possibility of mass production/storage and distribution during outbreaks, adequate rural region transportation, storage, and route of administration means [39,40]. Moreover, local community intervention acceptance is preferable, thus attending to socio-cultural traditions and “gachhi” professions derived from economic benefit preservation [35].

Any preventive or therapeutic strategy must meet all requirements for safety, effectiveness, cost, and availability. Amongst all the therapeutic agents investigated, m102.4 stands out as the most promising candidate for infection treatment; however, it has several important limitations as an antibody-based therapy not approved for human use [6,40,41,46]. On the other hand, a preventive approach might be much more appropriate to face the frequent NiV outbreaks in rural Bangladesh by attending to the health requirements and resource availability of the country. Multiple findings support this statement, including the difficulties for early diagnosis (most cases are diagnosed retrospectively), the minimal representation of NiV infection as a cause of encephalitis in Bangladesh, the strong limitation of supplying therapeutic agents, the scarce rural health infrastructure, the lack of infection control equipment for patient management, and the predominant seasonal pattern of NiV outbreaks [9,18,21]. Current evidence suggests focusing on preventive interventions in areas reporting elevated consumption of fresh palm sap, as well as the establishment of surveillance tools in order to trigger rapid outbreak responses. Interventions should emphasize both reduction of contaminated sap consumption through informative/educational strategies and physical barrier implementation during the sap harvesting process [23].

## 11. Future Directions

Several key features determine the epidemiological and public health contextualization of Nipah virus outbreaks in Bangladesh, including the high mortality and morbidity observed, the deficiency of health infrastructure in rural settings, and the intense delimitation of viral emergences to seasonal date palm sap consumption. Therefore, likely the most adequate strategy for tackling current necessities and means to confront NiV outbreaks in Bangladesh is a preventive approach focusing on risk areas (Nipah belt) during wintertime (Nipah season). Efforts should address the implementation of low-cost preventive interventions directed to block the viral interspecies transmission from *Pteropus* to humans, thus preventing the origin of human outbreaks. Both informative/educational interventions and physical barrier strategies, aiming to limit the consumption of contaminated sap, are considered crucial for this purpose.

Several lessons should be kept in mind about the NiV outbreak scenarios in South and South-East Asia:
Bangladesh, and specifically the Nipah belt, should be prioritized as the main target of anti-Nipah interventions because of the frequency and regularity of viral emergences. Human outbreaks in other Asiatic regions have depended on the presence of intermediate amplification hosts which facilitated the transmission to humans. For this to occur, the harmonization of several conditions were required, including the presence of a bat-to-amplification mammal interface and a subsequent amplification mammal-to-human interface. This defines an extraordinarily tough to predict temporo-spatial intersection, owing to the vast intercontinental geographic distribution of NiV-disseminating *Pteropus* spp. bats. Because of this, the rare viral emergences that may occur in other countries are very difficult to prevent. In contrast, Bangladesh’s frequent outbreaks are tightly linked to a specific risk exposure (date palm sap consumption); thus, an evident bat-to-human nexus determining the viral transmission is established. This also provides an obvious source of outbreak prevention by blocking bat-to-human transmission, which would minimize both regional and global outbreak-derived potential adversities. Further research and development of therapeutic agents is required due to the urgent need of treatment implementation in order to confront the extremely high mortality and morbidity of NiV human infection in endemic regions. A number of drugs and monoclonal antibodies have been developed and require future research in animal models and clinical trials for an eventual authorization for human use.Current economic investment in endemic regions should focus on low-cost, effective, and efficient preventive strategies able to interrupt interspecies transmission in the Nipah belt. The recommendations of the WHO and CEPI to assign priority to the development of therapeutic and preventive tools against NiV should serve to enhance the development and future application of effective measures in the countries with the greatest impact of the disease.A pragmatic consideration of the real epidemic/pandemic risk of novel emerging viruses is required, which relates to the need of infection contention assistance by the international community to certain regions with special risk of novel pathogen emergence and dissemination. Furthermore, as has been demonstrated with the COVID-19 pandemic, surveillance for potential pandemic agents, such as NiV, is essential for the implementation of early measures in local outbreaks or in broader circulation of the virus.The persistent human–wildlife interaction derived from environment modification by human activities will certainly promote the emergence of novel pathogens in the future, hence exposing large human populations to unpredictable threats.


## Figures and Tables

**Figure 1 viruses-16-00179-f001:**
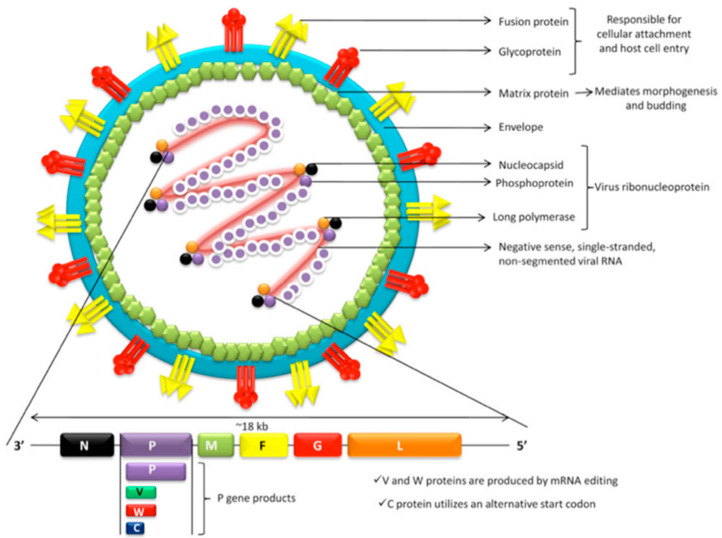
Nipah virus structure. Gene products. N, P, and L proteins constitute the viral ribonucleoprotein [17].

**Figure 2 viruses-16-00179-f002:**
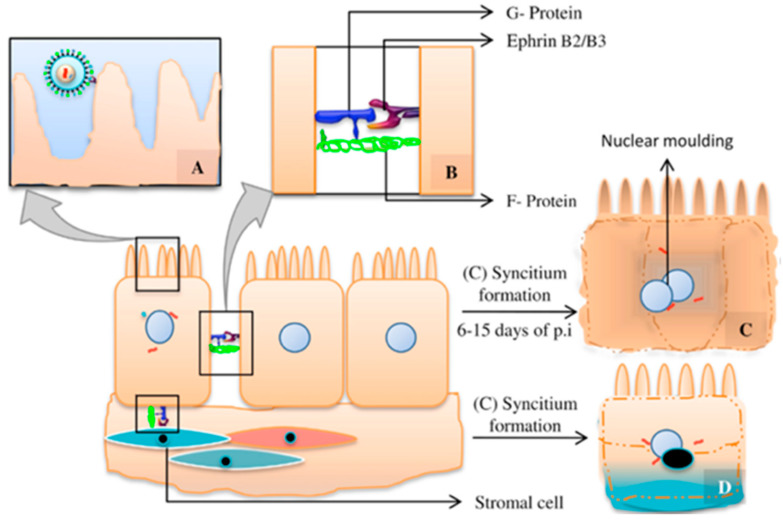
Cellular penetration and syncytia formation. (**A**) Viral attachment. (**B**) G glycoprotein–Ephrin B2/B3 receptor interaction and G glycoprotein–F glycoprotein interaction. (**C**) Syncytia formation by homologous epithelial cell fusion. (**D**) Syncytia formation by heterologous epithelial–stromal cell fusion [14].

**Figure 4 viruses-16-00179-f004:**
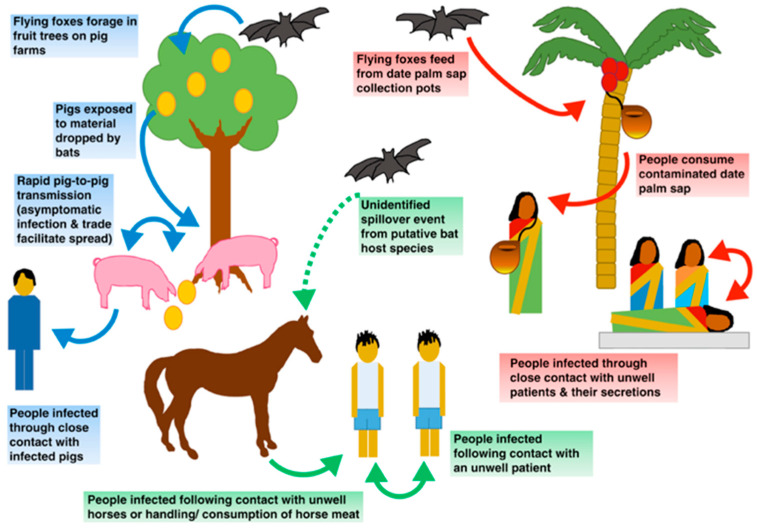
Nipah virus interspecies and interhuman transmission observed in different outbreaks. Malaysia-Singapore (blue), Bangladesh (red), and Philippines (green) [4].

**Figure 5 viruses-16-00179-f005:**
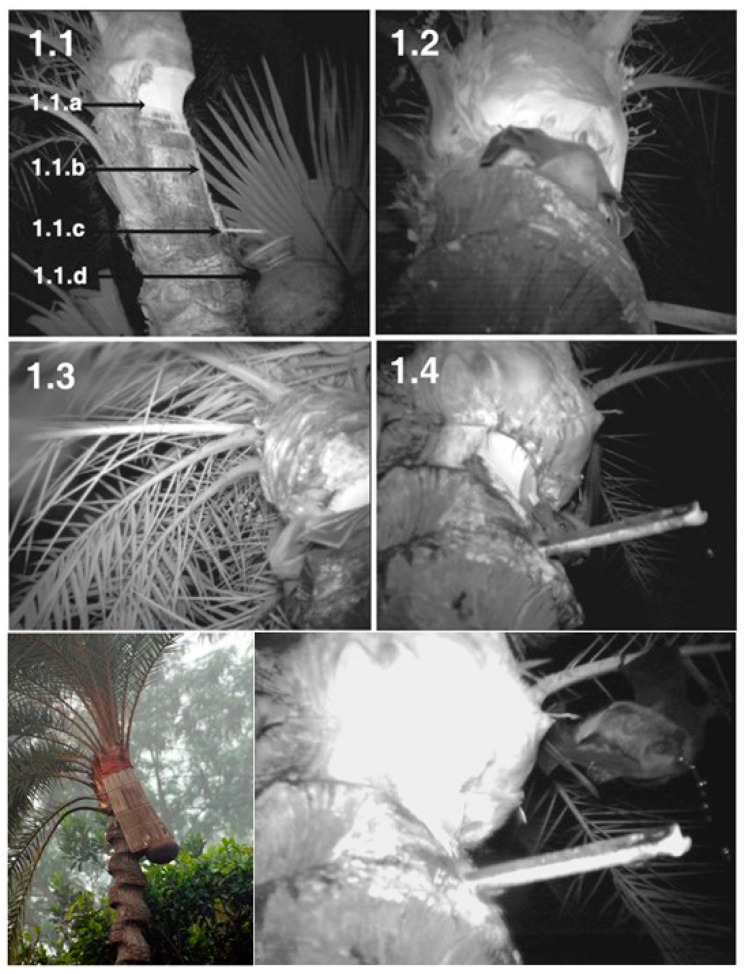
Date palm sap contamination by *Pteropus* spp. during harvesting process. (**1.1**): (a) fresh sap coming out from shaved bark surface, (b) fresh sap flow, (c) bamboo stick directing the fresh sap to a collection receptacle, and (d) collection receptacle. (**1.2**,**1.3**): *Pteropus* bat feeding from the fresh sap. (**1.4**): *Pteropus* bat licking shaved bark surface. (**Bottom left**): bamboo skirt covering sap circulation surfaces. (**Bottom right**): *Pteropus* bat urinating in proximity to sap circulation surfaces [51].

**Table 3 viruses-16-00179-t003:** Targeted transmission prevention messages [21].

For prevention of Nipah transmission through ingestion of raw date palm sap
Do not drink raw date palm sap: some bats carry Nipah virus and could contaminate raw sap during the collection process at night. Humans can get infected by consuming the raw date palm sap
Consuming boiled sap or molasses is safe
2. For prevention of Nipah transmission from person to person
Wash hands thoroughly with soap and water after coming in contact with patient Sleep in separate bed Maintain > 1 full-stretched arm distance (1 m or 3 feet) from patient Keep personal items of patient separately Wash used items of patient with soap and water, separately
3. For prevention of Nipah transmission at hospital setting
Admit all cases with fever and unconsciousness/convulsion/difficulty breathing to the isolation ward/facility in the hospital Use a mask and gloves during history taking, physical examination, sample collection, and other aspects of caregiving for suspected NiV cases Avoid unnecessary contact with suspected NiV cases Follow standard precautions for infection prevention at hospital setting Immediately report admission of a suspected NiV case to IEDCR ^1^ and relevant authority
4. Personal protection during care for Nipah patient
Use personal protection equipment
During history taking and physical examination, wear a surgical mask, surgical gloves, and a gown During specimen collection and other invasive procedures (such as nasopharyngeal suction, endotracheal intubation), wear an N95 mask, surgical gloves, and a gown
Hand hygiene
Wash hands with soap and water for at least 20 s, or Clean hands using 1–2 mL alcohol-based hand sanitizer (chlorhexidine or 70% alcohol hand sanitizers) after providing any care to patient
Use disposable items
Use disposable items while providing NG tube, oxygen mask, and endotracheal tube, or If disposable items are not available, reuse after sterilization by autoclave or 2% glutaraldehyde

^1^ IEDCR: Institute of Epidemiology, Disease Control and Research of Bangladesh.

## Data Availability

No new data were created or analyzed in this study. Data sharing is not applicable to this article.

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
