# Peer review of "Nipah Virus: A Multidimensional Update"

_viruses, 2024, doi:10.3390/v16020179_

Round 1

Reviewer 1 Report

Comments and Suggestions for Authors

This review offers an exceptionally comprehensive and highly detailed update of our present understanding of the emergence, spread and potential preventive measures for the control of Nipah virus.  The review encompasses pretty much the A-to-Z of NIpah virus from the emergence of the virus in the 1990’s with specific details relevant to the natural reservoir (various Pteropus species), the geographic distribution of the virus with emphasis on the so-called Nipah belt in central and northwest Bangladesh, the role of intermediate hosts (especially pigs and horses), virus pathology, tropism and immune evasion strategies, interspecies transmission, including human-human transmission, an evaluation of the potential efficacy of several therapeutic agents and culminates with a discussion of the most prudent strategies to be implemented in order to surveil and control the development of viral outbreaks.

There are no detected weaknesses in this review.  As someone with a longstanding interest in Nipah virus, this reviewer found the review highly illuminating with all aspects of the virus covered in one article, but most especially our understanding of animal-human transmission, which is not often discussed in such detail. 

In my opinion, this review is unique in its summation of our understanding of all aspects of Nipah virus.  Indeed, the only criticism I would have is in its title. My feeling is that the title shortchanges the comprehensive impact of the review.  In my opinion, the review accomplishes far more than ”focusing on therapeutic and preventive strategies”..  Thus, I would suggest either adding to the focus topics in the title or leaving it at “Nipah virus: a multidimensional update”.

Author Response

Response to academic editor and reviewers

Ref. No.: Viruses-2826420

Title: Nipah Virus: a multidimensional update

I would like to thank the reviewers and the editor for the time and effort to evaluate the earlier version of the manuscript. 

As a summary, the title has been modified as suggested by reviewer 1 and other minor changes have been addressed (highlighted in yellow in the manuscript), adding 3 new references, mainly to update treatment or vaccine information.

Reviewer 2 Report

Comments and Suggestions for Authors

The authors presented a review of the Nipah virus from various perspectives, including the viral outbreak, characterization, and reservoir, the disease infection and transmission process, and the countermeasures and strategies deployed. However, some current development in the field was not covered in the review. Please provide a more comprehensive discussion on the new publications in the related topics.

1.      Line 151, please edit this sentence, now it is not a complete sentence.

2.      Line 207, please consider adding labels to Figure 3.

3.      Line 243, please provide the full name of the agencies mentioned.

4.      8.2 Monoclonal antibodies: there are a few more potent antibodies discovered and described in publications in recent years, for example, antibody nAH1.3, 12B2, etc. Please add a discussion on the recent publications so that the review will be more informative to the readers.

5.      Line 421 vaccination: currently there are mRNA vaccines for the Nipah virus in clinical trials. Please review and provide related information.

6.      Please double-check the format, for example, line 335, line 369, line 440, etc.

7.      Line 467, please edit this sentence as now it is confusing.

8.      Please consider editing section 11, future directions, and elaborate on some of the bullet points that were lightly discussed. 

Comments on the Quality of English Language

Moderate editing of English language required
